# The White Sea: Available Data and Numerical Models

**Ilya Chernov** [1,*] and **Alexey Tolstikov** [2]

1   Institute of Applied Math Research of Karelian Research Centre of RAS, 185910 Petrozavodsk, Russia
2   Northern Water Problems Institute of Karelian Research Centre of RAS, 185910 Petrozavodsk, Russia;
    tolstikov@nwpi.krc.karelia.ru
*   Correspondence: IAChernov@yandex.ru; Tel.: +7-953-527-6261

**Abstract:** The White Sea is a small shallow semi-closed sea in the North-West of Russia. It is strongly affected by induced tides, so the tidal motion dominates in the sea. Sea ice is seasonal and the water salinity is less than in the neighbouring Barents sea due to strong river discharge. We review the sources of in-situ and satellite data that are available for the sea, and describe those few numerical models, together with the challenges that are faced. We focus on the large-scale circulation and thermohaline fields, but also cover sea ice, river runoff, and pelagic biogeochemical data.

**Keywords:** data; numerical modelling; simulation; White Sea; Arctic

## 1. Introduction

The White Sea is unique; it belongs completely to Russia, serves as gates to the Arctic, and it can be considered as a model of the Arctic [1]. In this review, we describe sources of in situ and remote sensing data, which were collected for the sea, available reanalysis data, and comprehensive numerical models that are maintained today.

The White Sea has been playing an important role for Russia for ages. As it is an inner sea, its resources were initially mostly used by the people who lived at its coast. This group of people is called pomors; they were rather isolated, built special ships, developed fishing gear and methods of hunting the sea animals, and learned to navigate in the harsh boreal conditions. This knowledge allowed pomors to travel out of the White Sea far to the north, up to the Spitsbergen (also known as Svalbard) and Novaya Zemlya archipelago in the Kara Sea. Therefore, the White Sea was the starting point of exploring the Arctic, as far as 500 years ago.

During the Soviet time, the White Sea-Baltic Canal was built and cargo shipping along the Northern Sea Route was launched. The White Sea became an important transport hub and it also remained important for fishing, essential for the economy of the country. Besides the fish, algae and molluscs were also harvested. Deposits of various metals, including precious, and diamonds were discovered.

The amount of data accumulated for the White Sea is rather decent, but there is a lack of coupled comprehensive numerical models adjusted for the sea. A survey of the data collections and sources, as well as of numerical models for the sea, is the subject of this paper (see also [2]).

To face various challenges, data on many quantities are necessary:

- hydrological, including water temperature, salinity, and density; sea-ice thickness, amount of snow on the ice, sea-ice concentration; current and sea-ice drift velocity, sea level; and,
- hydrobiological, including biomass of different types of plankton, productivity, and chlorophyll content.

Data can be obtained either in situ (from vessels, autonomous gliders or buoys) or by remote sensors [3]. Some estimates can be obtained by numerical models. However, in order to verify and

tune a model, as well as for initial and boundary data, measured data are still necessary. Figure 1 shows the White Sea bathymetry according to ETOPO (https://www.ngdc.noaa.gov/mgg/global/) with main rivers and main parts of the Sea.

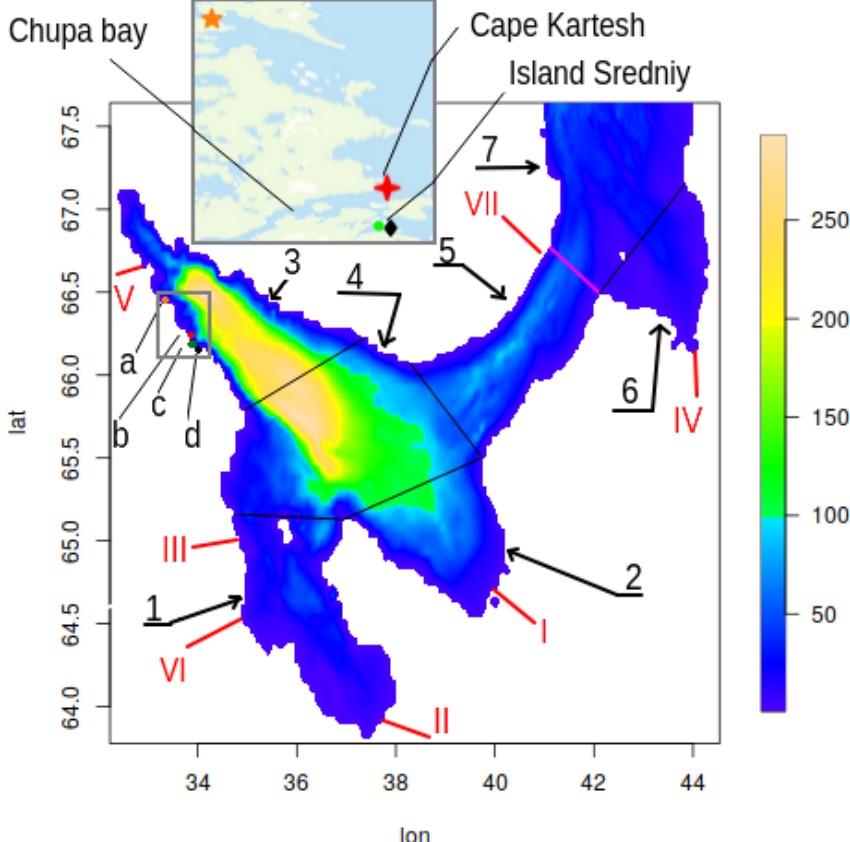

**Figure 1.** The White Sea bathymetry. The colour shows the depth in metres. Black lines are boundaries of the bays marked by arabic numbers: 1—Onega Bay, 2—Dvinskiy Bay, 3—Kandalakshskiy Bay, 4—Bassein, 5—Gorlo, 6—Mezenskiy Bay, 7—Voronka. The pink line separates Voronka and Gorlo and is also the open boundary of the numerical White Sea model by E. Semenov. Red roman numbers mark the main rivers: I—Northern Dvina, II—Onega, III—Kem, IV—Mezen, V—Kovda, VI—Nizhniy Vyg (part of the White Sea-Baltic Canal), VII—Ponoy. Letters a-d mark positions of biological stations: a—Moscow State University, b—Zoological Institute of the Russian Academy of Sciences, c—Saint-Petersburg University, d—Kazan University; the incut shows part of the Kandalakshskiy Bay with the stations in detail.

## 2. In Situ Data

Obtaining in situ data in the White Sea is not easy. Regular seasonal measurements in all areas of the sea are not carried out due to various reasons, although the White Sea is internal and logistically convenient enough for organizing expeditions. Nevertheless, there are either fragmentary observations obtained in expeditions, or permanent long-term observations carried out at biological stations, and so are concentrated on a relatively small part of the south-west coast of the Kandalakshskiy Bay (Figure 1). Besides, measurements in expeditions are carried out with different equipment, irregularly in time (usually from June to August), at different depth levels, and they are quite sporadic spatially. Hence, it is difficult to extrapolate such data and compare measurement series with each other. There is more data for the sea surface than for the water column.

However, this does not mean that there are no acceptable resources for modelling circulation and biogeochemical dynamics in the White Sea. Regular observations in the water area are carried

out by the institutions of the Federal Service for Hydrometeorology and Environmental Monitoring (Roshydromet), Federal Fishing Agency, as well as the Ministry of Transport. The Ministry of Defense is also carrying out work. However, for obvious reasons, access to such data is strictly regulated for scientific organizations.

Of course, most of the in situ data are obtained by the Federal Service for Hydrometeorology (Roshydromet) during expeditions. Besides water temperature and salinity, data on biogeochemical tracers in the water column are collected at different times of the year and include the so-called secular sections. Roshydromet conducts observations on a standard grid of hydrometeorological stations [4] according to standard methods. These measurements are regular, long-term, and well-structured. When modelling thermohydrodynamic processes, information about the ice regime, characteristics of the runoff of rivers flowing into the sea, and meteorological data is important. These observations are also performed at standard stations. The main obstacle to the use of Roshydromet data is that access to them is commercial, even at the official request of scientific institutions. Only a part of these materials can be obtained free of charge through the Obninsk World Data Centre (http://meteo.ru/english/index.php) , which is part of the Roshydromet structure and is its main institution for the accumulation, storage, and processing of hydrometeorological information. This organization publishes regional directories, catalogues, creates open electronic resources, but access to a significant part of the data is also not free.

The data of the Zoological Institute of the Russian Academy of Sciences obtained at the biological station near Cape Kartesh (http://www.zin.ru/kartesh/) (see Figure 1) are useful for simulating the White Sea. Measurements are carried out at the ten-day station D-1 (66°20.230' N; 33°38.972' E) for several decades, every 10 days from May to October. Temperature and salinity are measured in the water column at several depth levels from 0 to 65 m (since 2010 continuous profiling of the water column with a CTD probe is done), concentrations of biogenic elements, abundance of zooplankton, and, from time to time, phytoplankton are measured. In winter, observations are also conducted here, but much less frequently (once a month or two, depending on the ice conditions). During summer the deep-water station in the central part of the Kandalakshskiy Bay is available. All these data are kept in an electronic database (http://www.st.nmfs.noaa.gov/copepod/time-series/ru-10101/) Access is possible through an agreement with the Institute. Some data have been obtained in a joint project with the American National Oceanic and Atmospheric Administration (http://www.noaa.gov/) (NOAA) and are now available in electronic form on the NOAA website (https://www.nodc.noaa.gov/OC5/WH_SEA/WWW/HTML/atlas.html ).

Educational stations of universities make a significant contribution to the knowledge about the White Sea gathering in situ data; they are also located on the south-west coast of the Kandalakshskiy Bay (see Figure 1). The biological station (http://www.wsbs-msu.ru/) of Moscow State University is located north to the Chupa Bay (66°34' N, 33°08' E). The marine biological station (http://spbu.ru) of St. Petersburg State University is at 66°17'21" N, 33°39'41" E and the station (http://kpfu.ru) of the Kazan Federal University is located on the island Sredniy. Hydrophysical and biogeochemical parameters are also measured. In addition to measurements done at the stations, all three universities organize expeditions in the White Sea (for example, [5,6]), and contribute databases of hydrological and biogeochemical data. Most of these databases are copyrighted.

Yearly expeditions are also carried out by P.P. Shirshov's Oceanology Institute (https://ocean.ru/) of Russian Academy of Sciences and its Saint-Petersburg branch [7–10].

Student training expeditions by Russian State Hydrometeorological University (http://www.rshu.ru/) in the White Sea collected a significant multi-year amount of data on temperature, salinity, and currents of the sea.

The data of the Murmansk Marine Biological Institute (http://www.mmbi.info/) of the Kola Science Centre of the Russian Academy of Sciences (MMBI KSC RAS) contain information on ship in situ measurements of temperature and salinity, starting from 1891, in different seasons and different years. Besides, much expedition data on ecosystem indicators (biogenic elements, distribution of

chlorophyll *a*, etc.) have been assembled here. The joint project of MMBI KSC RAS and NOAA "Climatic Atlas of the Arctic Seas 2004: Part I. Database of the Barents, Kara, Laptev, and White Seas—Oceanography and Marine Biology" (https://permanent.fdlp.gov/lps118727/english58.pdf) presents temperature and salinity maps of the White and Barents Seas together; this is important for setting the boundary conditions in numerical models.

Expedition data of the Northern Water Problems Institute (http://water.krc.karelia.ru/) of the Karelian Research Centre of the Russian Academy of Sciences, obtained in joint research projects with many organizations, are also useful for tuning numerical models. The following research programs of the last twenty years are worth mentioning: Federal Research Program "World Ocean", INCO-Copernicus, INTAS, Russian Foundation for Basic Research projects. The data on the temperature and salinity of the White Sea water and concentration of biogen matter and chlorophyll *a* at various depth levels in different regions from the year 2000 up to today has been collected in a database; most measurements were made in ice-free periods (although some data was collected during winter). All these data are contained in an off-line database [11] and are available on request.

Data of the Institute of Northern Environmental Problems (http://fciarctic.ru/index.php?page=iepn) of the Ural Branch of the Russian Academy of Sciences in Arkhangelsk are of special interest. They have accumulated data on thermohydrodynamics, hydrochemistry, and ecology of the White Sea. Measurements on the mouth of the river Northern Dvina and Dvinskiy Bay as a whole, as well as Onega Bay, are added yearly.

The N.M. Knipovich Northern Polar Scientific Research Institute (http://www.pinro.ru/15/index.php/structure/sevpinro#) of Marine Fisheries and Oceanography, an institution of the Federal Agency for Fishery in Arkhangelsk, has accumulated data on the catches of algae, fish, and marine mammals in the White Sea. A significant part of the expeditionary work carried out by this Institute refers to monitoring, so it is done regularly. However, access to these materials for other organizations is strictly regulated.

Data on river runoff into the White Sea are possessed by Roshydromet (http://www.r-arcticnet.sr.unh.edu/v4.0/index.html); among the free sources, the most convenient seems to be R-ArcticNET: it contains data on river discharge, including the Northern Dvina, Onega, Mezen, Kovda, Ponoy, White Sea-Baltic Canal starting from 1999 (earlier for some rivers).

Bathymetry is necessary for the modelling of hydrophysical and biogeochemical processes in a sea. A bathymetric map of the White Sea has been created in a joint Russian-Finnish project [12]. Another map was created by the morphogenetic analysis of the White Sea bottom relief [13]. Often, the free ETOPO (Global Digital Elevation Model) resource by NOAA is used (https://www.ngdc.noaa.gov/mgg/global/). These datasets give smaller area and volume of the Sea (see Table 1), when compared to the area given in the sailing directions (http://rivermaps.ru/doc/beloe/beloe-1.htm) and volume from [14] (note that [15] gives much higher volume: 6000 km$^3$). We calculated the Sea area while using the 1 to $10^6$ map and a GIS and obtained the same value (91,000 km$^2$).

**Table 1.** Sea depth, area, and volume according to different bathymetry data sets. Relative error is with respect to the sailing directions values.

| Dataset | max, m | Volume, km$^3$ | Area, km$^2$ | V Error, % | A Error |
|---|---|---|---|---|---|
| INCO-Copernicus [12] | 333 | 4362 | 71,074 | 19% | 22% |
| Digital terrain model [13] | 293 | 4484 | 76,558 | 17% | 16% |
| ETOPO | 233 | 4466 | 80,802 | 17% | 11% |
| Sailinig directions and [14] | 350 | 5400 | 91,000 | — | — |

Atlases, as collections of data maps, are particularly useful for assessing the adequacy of numerical models. Most atlases of the White Sea [16–22] contain comprehensive information on the Sea. The most detailed hydrometeorological dataset is in the electronic atlas of the ESIMO project "Climate of the seas of Russia and key regions of the World Ocean. White Sea" (http://www.esimo.ru/atlas/Beloe/1

_1.html) . A new comprehensive atlas of the White Sea and the watershed has been developed [23,24]. It includes data from most of the sources described in this section. The information included in the atlas is stored at the NWPI KarRC RAS and it is available upon request.

Spatial coverage is uneven because of the fact that some parts of the sea are closed for expeditions. This can change from year-to-year. Usually, Kandalakshskiy or/and Dvinskiy bays are closed, while Onega Bay is available.

## 3. Remote Sensing Data

Satellites provide a significant amount of the White Sea data. For example, the online resource of Oceanology Institute (http://optics.ocean.ru) by O.V. Kopelevich et al., provides information on the distribution of surface temperature, suspended and yellow matter, backscatter coefficient, chlorophyll *a* for the warm season. Some of the satellite data (distribution of temperature and chlorophyll *a*) were obtained in 2009 during joint work by the NWPI (http://water.krc.karelia.ru/) of Karelian Research Centre of RAS and the Nansen's International Centre for the Environment and Remote Sensing (http://www.niersc.spb.ru/home.html). All of these measurements are contained in the database [11]. A survey of ocean remote sensing in USSR and Russia is given in [25].

Due to climatic conditions, data are available for May–September [26]. Temperature measurements are accurate enough if the sky is clear. However, ice concentration can be estimated well in winter, provided that there are no clouds. With chlorophyll, the remote sensor data reliability is the most questionable. Standard algorithms may produce large errors due to strong river discharge, turbid water, and an abundance of optically active matter. This is typical for the Arctic region, but even more important for the White sea (4% of the volume discharged by rivers yearly, the euphotic layer is 10–15 m). The regional algorithm has been developed [26], but it is not reliable enough due to the low number of in situ control measurements: 68 pairs of concurrent in situ and satellite. In [27], a very high variability of chlorophyll *a* is reported (see Table 2 there). For instance, for June 2008 the range was from 0.27 to 9.17, for 80 measurements. Also, the station measurement (Figure 1 in the paper) reports 2–2.5 µg/L, while the satellite data in the neighbourhood is 1.5–1.7 and 1.7–2. However, the data maps (http://optics.ocean.ru) give no less than 2 µg/L in most of the sea. The paper [5] reports other values for the chlorophyll (µg/L, see Table 3): 0.727 for the Onega Bay (10 stations) and 0.702 for the Chupa Bay (5 stations). Additionally, chlorophyll *a* concentration was between 1.9 and 9.8 µg/L in the Dvinskiy Bay and between 0.54 and 1.16 µg/L in the Onega Bay in 1991, according to the book [4]. The regional algorithm uses averaging over $2 \times 2$ squares and long time periods (one month or even a year), which relies on the mean of instant measurements of concentration. However, the atlas (http://optics.ocean.ru) is already a decent result and the best possible to be offered for such a complex region as the White Sea. The qualitative distribution of chlorophyll during the productive season is reproduced well.

Many satellite data sources are free and open, and, therefore, rather available: e.g., NOAA (https://www.nnvl.noaa.gov/view/globaldata.html), NASA (https://oceancolor.gsfc.nasa.gov), and Scanex (http://new.scanex.ru/) . Let us mention the research centre for space hydrometeorology "Planeta" (http://planet.iitp.ru) and the satellite data centre ESIMO (mentioned above (http://www.esimo.ru/atlas/Beloe/1_1.html) . For the ice situation in the White Sea, the "Multimaps" (https://multimaps.ru) is convenient. It gets data from satellites Terra, Aqua, and Suomi NPP, for a chosen date. A bathymetry map and animated wind, wave, and cloud maps are offered, as well as a short-term forecast of the state of the water surface and weather.

Reference data of the Russian Maritime Register of Shipping [28] are useful for setting up mathematical models. It includes typical wind speeds and wave heights for different seasons, expected frequencies of extreme events for different periods, including 30-year and secular. Note that the NCEP/NCAR reanalysis is used in addition to the collected data.

The NCEP/NCAR reanalysis is NOAA (National centre for Environmental Forecasting and National Centre for Atmospheric Research) meteorological data, for the entire globe on a regular

grid, starting from 1948, with the six-hour time step. This dataset (atmospheric pressure fields, wind, air temperature, etc.) is updated monthly and it is freely available (https://rda.ucar.edu/datasets/ds090.0/#!description) to researchers. NCEP/NCAR data are one of the most commonly used by numerical sea models. A summary of resources providing reanalysis data is available on the NCAR/UCAR (https://climatedataguide.ucar.edu/climate-data/atmospheric-reanalysis-overview-comparison-tables) and the University of Hamburg (http://icdc.cen.uni-hamburg.de/projekte/easy-init/easy-init-ocean.html) pages.

Another drawback of the remote sensing is that it only provides information on the sea surface.

## 4. Online Resources

In this section, we briefly describe the available free internet resources of the White Sea data (although most of them also offer data on other seas and other kinds of data). Some of the resources are based on the data sources described in the previous sections.

The web resource (https://meteoinfo.ru/ocean) of the Hydrometeorological Centre of Russia offers the today sea-surface temperature and ice concentration for several seas of Russia, including the White Sea; the ten-day average ocean-surface temperature of the global ocean; sea-ice concentration for the Arctic and the Antarctic; and, monthly average climatic sea-surface temperature for the global ocean.

The reanalysis data from The European Centre for Medium-Range Weather Forecasts (ECMWF) (https://www.ecmwf.int/en/forecasts/datasets/browse-reanalysis-datasets). This is the European counterpart of NCEP and it sometimes gives more reliable predictions for Arctic regions [29].

The ESIMO web resource has been already mentioned (http://www.esimo.ru/atlas/Beloe/1_1.html): it is an electronic atlas that contains climatic data of the White Sea: water and air temperature, water salinity and density, sea level, wind speed, waves, concentration of oxygen in water, and sound speed. It also contains information on pollution (http://esimo.oceanography.ru/esp2/index/index/esp_id/12/section_id/12/menu_id/4346) of the White Sea.

The data collection of the Zoological Institute of the Russian Academy of Sciences is available on the NOAA website (https://www.nodc.noaa.gov/OC5/WH_SEA/index1.html) and it contains water temperature and salinity and zooplankton measured from 1963 to 1998 on a biological station.

The "Ocean color" resource (http://optics.ocean.ru) offers maps of sea-surface water temperature, ice and chlorophyll *a* concentration, obtained via the satellite remote sensors.

Reference data (https://ohranatruda.ru/ot_biblio/norma/563721/) on wind and waves of the Bering and White seas are available; the White Sea starts from page 371; page 373 contains useful information on the monthly-mean sea-ice distribution in the White Sea.

The "Planeta" resource (mentioned above (http://planet.iitp.ru) offers data, mostly satellite, on the wind, ice, and surface temperature of the White Sea.

The Aphrodite (https://www.hetwaterleven.com/afrodita) resource is aimed at collecting data on the abiotic conditions of the White Sea. At the moment, only the water temperature is measured in the water column on one biological station by several sensors. Data are available for years 2016 up to now.

## 5. Comprehensive Numerical Models

Numerical models are used for simulating the sea circulation and estimating various quantities, including those that are hard or impossible to measure in situ. Numerical experiments allow for simulating consequences of events that are rare or possible, but have not happened yet; for example, the influence of an ice-free winter, drastic change in river runoff or concentration of biogenic matter in river water, etc.

Numerical models provide discrete grid values, with a given resolution. The horizontal grid resolution for small seas, including the White Sea, vary between several kilometres (at best, several hundred meters, but we do not know such models for the whole White Sea). This creates a

certain contradiction between model predictions of highly varying quantities (such as, for example, the concentration of chlorophyll *a*) and in situ measurements, which complicates comparison. For example, samples in close positions and taken with a short time interval can give an order of magnitude discrepancy [27], while, from the point of view of the model, these samples correspond to the same node of the space-time grid. Local features of the hydrological regime (a river flowing into the sea, intense currents in the strait between small islands, fast ice, etc.) can significantly affect the local values of many quantities, but they are completely transparent for a model, which is unable to resolve objects of this scale.

Initial and boundary values are required for a simulation. Initial fields are needed once, but on the entire 3D grid. This presents a significant difficulty, known as the "initial data problem". Sometimes it is necessary to apply computationally expensive methods of assimilation of observational data [30]. Fortunately, this problem is less acute for the White Sea due to the dominant influence of strong induced tides [14,31,32]. Tidal dynamics determine the stable pattern of circulation, temperature, and salinity, as well as passive tracers (including biogeochemical concentrations). Numerical experiments have confirmed this statement: after several months of model time, the differences in the distribution of the initial fields become barely noticeable. Besides, the White Sea is ice-free in summer and, therefore, the characteristics of ice are determined each year only by the state of the sea and they do not depend on the initial distribution.

Therefore, the simulation of the White Sea needs just climatic average values for the whole sea or its main parts, at several depths. These values can be estimated from the sources discussed above.

Let us consider the boundary conditions. The only open boundary for the White Sea is the border with the Barents Sea; however, temperature and concentrations of biogeochemical substances in rivers (values at river mouths) are necessary. Due to relatively shallow depths at the river mouths, estimates of values on the surface are sufficient. In contrast to the initial data, the boundary values are required in a relatively small number of points on the spatial grid, but at every moment in time. A reasonable compromise is to use monthly averages for a typical year (for example, multiyear averages). The problem of assessing the sensitivity of the model to boundary values should be investigated.

Precipitation and evaporation are close to balance [4] and theybdo not significantly change the amount of water. The river runoff is significant, about 4% of the sea volume per year, and thus the water outflow is more than inflow for the White Sea [32]. So, we can assume that boundary values would not drastically influence the Sea. To test this assumption, we increased the boundary value of water temperature by 2.0 °C and compared the daily average surface temperature of the whole sea and separate bays with the test simulation. The difference was, at most, 0.3 °C for the whole Sea, while the maximum difference was near the boundary: in Voronka, and it was at most 1.2 °C.

Let us list the challenges that are faced by a model. The sea waters have high energy due to strong induced tides. Maximum tidal speed can be very high (250 cm/s as compared to 10–15 cm/s for the mean current speed), and the average speed at the surface is also higher compared to neighbouring seas. Additionally, the sea is small, so the spatial step is low, and, therefore, the time step needs to be small. Accordingly, the numerical model is more demanding. The baroclinic Rossby radius is 1–4 km: this also demands high grid resolution. The Sea is shallow, so the bottom influences the currents significantly and exchange of matter between benthic and pelagic ecosystems is rather quick. Additionally, the depth is quite variable, with very shallow bays (10–20 m) and a deep place (340 m) in the middle. Strong river load contributes to the stable stratification, while tidal mixing destroys it, and it is very important to carefully address this matter.

There are very few comprehensive modern numerical models of the White Sea. There are only two models maintained now (the model by prof. I.A. Neelov and O.P. Savchuk [14] is not used for the White Sea now). One is JASMINE, as discussed below, the other is the Operational model of monitoring of the White Sea [32]. The Operational model was created and it is being developed by prof. E. Semenov in the institute of Oceanology of RAS. It is a finite-difference model for the Primitive

hydrostatic equations. Just a part of the White Sea is considered: the boundary between Gorlo and Voronka serves as the open boundary of the model domain (see Figure 1, where this boundary is also shown). The model is operational, i.e., it is used to forecast several days or weeks of sea dynamics. Therefore, it is only used for ice-free periods and, to our knowledge, does not contain a sea-ice submodel. Recent applications of the model are described, e.g., in [32–35]. The model is reported to reproduce the most important eddies and, to our knowledge, has never been compared to temperature and salinity measurements. The operational version [33,35] calculates the sea dynamics for several days and assimilates observed data.

The numerical model JASMINE is based on the Finite-Element Model of the Arctic Ocean (FEMAO [36]); it has been developed in the Karelian Research Centre and it is a numerical model of the White Sea. It is a partially finite-element, partially finite-volume model, with the usual Boussinesq and hydrostatic approximations. The sea hydrothermodynamics model is coupled with the comprehensive model of sea ice and the model of marine ecology BFM [37] (Biogeochemical Flux Model). BFM describes matter fluxes between groups of plankton organisms, particulate and dissolved matter, benthic sources, and sea-ice ecosystems ([38]). At the moment, there are two stable configurations: 8 km version [39] and 3 km version. The numerical experiment presented above was performed on both of the configurations. The grid of the first setup is $50 \times 50$ horizontal points and 16 non-equidistant depth levels; the second setup uses $200 \times 200$ points and 31 depth levels 5 m apart plus 15 depth levels 10 m apart. As [39] reports, the model reproduces sea-surface temperature and salinity well; however, there is still no statistical comparison. The mixing in the water column is overestimated (for the deep parts of the Sea), so the deep-water temperature in the model varies during a year more than it really does: it is always negative, but it can reach 1 °C in the model. The same is with deep-water salinity. Salinity measured near a river mouth may be much less than the model prediction. The reason is the resolution: the model considers a grid node (3 km×3 km) as a point, while the station may be in fresh water when the runoff increases in spring. The ecosystem submodel needs to be further adjusted. Concentrations of nutrients, i.e., nitrates and phosphates, are overestimated. Chlorophyll is hard to compare because of large variability, but it seems to be underestimated: the average values are approximately 1 μg/L.

## 6. Conclusions

In situ data gathered in expeditions remain the main source of knowledge regarding the White Sea. They are vital for verifying numerical models and for boundary and initial data. Remote sensor (satellite) data are not always available due to cloudiness that prevails over the White Sea for half a year. Using the data for chlorophyll is questionable due to the necessity of improving regional algorithms and other complications discussed above. Though, such data is useful for qualitative comparison. The only numerical model of the White Sea with a sea-ice submodel and suitable for simulating consistent thermohydrodynamic and biogeochemical processes is still being improved.

**Author Contributions:** Conceptualization, A.T.; Models, I.C.; Software, I.C.; Validation, A.T.; Investigation, A.T. and I.C.; Writing—Original Draft Preparation, A.T.; Writing—Review & Editing, I.C. and A.T.; Visualization, I.C. All authors have read and agreed to the published version of the manuscript.

**Funding:** The modelling study was carried out under state order to the Karelian Research Centre of the Russian Academy of Sciences (Institute of Applied Mathematical Research and Northern Water Problems Institute of KarRC RAS, State registration number AAAA-A18-118032290034-5).

**Conflicts of Interest:** The authors declare no conflict of interest.

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
