# Peer review of "The White Sea: Available Data and Numerical Models"

_geosciences, doi:10.3390/geosciences10110463_

Round 1

Reviewer 1 Report

The authors are trying to list the sources of available in-situ and satellite data for the White Sea. I think it is a good idea to have such an article to help researchers find the data sources for their study. However, in my opinion, the authors need to be more specific when it comes to numerical models and remote sensing data.

The authors have concluded that remote sensing data is questionable and needs improvement, but they have not supported this conclusion well. The remote sensing data in the Arctic circle is not that good; it is a known fact. Therefore, it will be better to give some statistics showing how bad the data are. Similarly, they have claimed that numerical models are not that accurate. It is true, and a well-known fact. But it will make more sense if people can know how bad they are. Understanding the efficacy of the data and algorithms will help the researchers decide whether they can use them in their study.

In brief, the article lacks a firm conclusion. I would recommend this article for publication if the authors can add some statistics to their conclusion. This article is a review and should help the researchers to make a more informed decision, and without proper quantification of how good the data or models are, it will be difficult.

Minor comments:

Please note that, in the following text, I am abbreviating Page as P and Line as L, which means P1L1 means Page one Line 1.

  1. In P1L9, the authors have stated that the White sea belongs completely to Russia. Then in P1L28, authors again state that “The White Sea completely belongs to Russia”. I don’t think that it needs to be said again and again. You said it once, I believe you.
  2. In P2L46, the authors say that measurements are carried out by different instruments at different times and depths. But does it pose a problem? If all the instruments are recording, let’s say Salinity, and all of them are calibrated, I think they should be comparable.
  3. In P5L126, “Both these datasets give 20% smaller area and volume of the sea.” Here, “Both these” is not appropriately defined. The authors state about three datasets, Russian-Finnish project, morphogenic analysis and the ETOPO, which of these datasets give 20% smaller area. I t will be better if the authors can provide more precise numbers, like ETOPO, give 20% smaller area and X% smaller volume.
  4. In P5L136, is it necessary to give the title of the article?

Author Response

We are very much grateful to the Reviewer for valuable comments!

The authors are trying to list the sources of available in-situ and satellite data for the White Sea. I think it is a good idea to have such an article to help researchers find the data sources for their study. However, in my opinion, the authors need to be more specific when it comes to numerical models and remote sensing data. The authors have concluded that remote sensing data is questionable and needs improvement, but they have not supported this conclusion well. The remote sensing data in the Arctic circle is not that good; it is a known fact. Therefore, it will be better to give some statistics showing how bad the data are.

Reply: we agree and have added some discussion and numbers

Similarly, they have claimed that numerical models are not that accurate. It is true, and a well-known fact. But it will make more sense if people can know how bad they are.

Reply: we agree; it is hard to compare somebody else's model with data. We added a discussion on the other model and some comparison for the one we are developing. Understanding the efficacy of the data and algorithms will help the researchers decide whether they can use them in their study. In brief, the article lacks a firm conclusion.

Reply: we agree and tried to make the conclusion better and more informative.

I would recommend this article for publication if the authors can add some statistics to their conclusion. This article is a review and should help the researchers to make a more informed decision, and without proper quantification of how good the data or models are, it will be difficult.

Minor comments:

Please note that, in the following text, I am abbreviating Page as P and Line as L, which means P1L1 means Page one Line 1.

In P1L9, the authors have stated that the White sea belongs completely to Russia. Then in P1L28, authors again state that “The White Sea completely belongs to Russia”. I don’t think that it needs to be said again and again. You said it once, I believe you.

Reply: We are sorry for this dublication; the first was an introductory sentence, to attract attention, the second was a fact from a fact list. We have changed the text, removing the sentence.

In P2L46, the authors say that measurements are carried out by different instruments at different times and depths. But does it pose a problem? If all the instruments are recording, let’s say Salinity, and all of them are calibrated, I think they should be comparable.

Reply: Some of the data obtained in oceanological expeditions and biological stations long ago (15--20 years ago) were obtained by rather simple old devices (made in the USSR) instead of modern probes; precision drastically depended on the scill of the measuring person. Other data were obtained by the probes SeaBird, Sea and Sun, CastAway; each probe has its own precision and error. Of course, these measurements are more representative and can be compared. However, measrements made at different depth levels and during different times of year are hard to compare. Pieces of data got for one region of the sea can not be extrapolated to the whole sea because different parts of the sea are quite different. We agree that variation of the salinity measurements is low. What we would like to declare is that measurements are quite sporadic spatially and have been obtained at different depths.

We added a sentence to make the idea a little clearer.

In P5L126, “Both these datasets give 20% smaller area and volume of the sea.” Here, Both these” is not appropriately defined. The authors state about three datasets, Russian-Finnish project, morphogenic analysis and the ETOPO, which of these datasets give 20% smaller area. It will be better if the authors can provide more precise numbers, like ETOPO, give 20% smaller area and X% smaller volume.

Reply: We added a table with numbers and a little description.

In P5L136, is it necessary to give the title of the article?

Reply: We reduced the line, removing the title.

Reviewer 2 Report

Chernov and Tolstikov: The White Sea: Available Data and Numerical Models

This review of data resources and numerical models for the White Sea is short and concise. My comments below are mainly concerning lack of references to web sites and including all names mentioned in the text into the map in Figure 1. Also, the text needs to be language cleaned by a person familiar with the English language. I have mentioned quite a few word, but I am also not fluent in the English language.

Comments referring to line numbers:

Line 10-11: common terminology use ‘remote sensing data’

Line 20: mark White Sea-Baltic Canal in Figure 1, and change ‘Canal’ to ‘Channel’ like in line 121.

Line 22: change ‘fishing object’ to ‘fishing area’?

Line 23: change ‘extracted’ to ‘gathered’?

Line 25: ‘The White Sea…’ is a repetition of the first sentence. Reconsider if this is really an important statement.

Line 26: Add reference to which bathymetric data gives smaller values, and consider giving the numbers.

Line 36 ‘remote sensing’

Line 41: delete ‘of’

Line 47: by ‘horizons’ I think you mean ‘depth levels’

Line 50: change ‘processes’ to ‘purposes’?

Line 69: add Cape Kartesh to the map, Figure 1.

Figure 1: all place names written in the text should be indicated in the map. I find these in the text: ’White Sea Baltic Channel’(l.20, l.121), ‘Cape Kartesh’(l.69), ‘Karelian coast’(l.81), ‘Chupa bay’(l.82,84), ’Ponoy’(l.121), ‘Voronka’(l.230), ‘Gorlo Bassein’(l.244’). Also for b -Zoological Institute you should probably add ‘Russian Academy of Science’.

Line 81: add Karelian coast to the map, Figure 1

Line 82; add Chupa Bay to the map, Figure 1.

Line 93: add ‘(CMMBI KSC RAS)’ after the full name, so the reader understands what you refer to in line 96.

Line 95: use a standard type of writing for ‘chlorophyll-a’ throughout the manuscript. ‘a’ written in italic is most common.

Line 95: change ‘accumulated’ to ‘assembled’?

Line 96: add a reference or web site to the “Climate Atlas of the Arctic Seas 2004: Part I…”

Line 100: change ‘Expeditionary’ to Expedition’?

Line 105: ‘chlorophyll-a’ with a in italic.

Line 111: find a better word than ‘contributed’

Line 116: change ‘fishing’ to ‘catches’?

Line 117: delete ‘on’.

Line 121: add position of also the Channel.

Line 126-127: ‘20% smaller’ than what? Combine the information here with your information in lines 25-27.

Line 131: ‘At present’? The references 21 and 22 are from 2014 and 2017, and thus a few years old. Are there web sites or newer updates you could add?

Line 135: ‘Remote sensing data’

Line 139-140: ‘chlorophyll-a’, sea comment to line 95.

Line 140-141: any web site or reference you could add about this joint work? Or at least to ‘NWPI…’ and ‘Nansen’s International…’

Line 146: add web site 18 for ‘ESIMO’?

Line 163: I am surprised you do not mention the reanalysis data from The European Centre for Medium-Range Weather Forecasts (ECMWF): https://www.ecmwf.int/en/forecasts/datasets/browse-reanalysis-datasets

This is the European counterpart of NCEP, and sometimes gives more reliable predictions for Arctic regions, see e.g. Fan et al, 2008: A one-year experimental Arctic reanalysis and comparisons with ERA-40 and NCEP/NCAR reanalyses. GRL, 35, L19811 https://agupubs.onlinelibrary.wiley.com/doi/pdf/10.1029/2008GL035110

line 172: is ‘ESIMO web resource’ the web site 18 mentioned in line 131?

Line 172: please add web site reference to ESIMO also here.

Line 174: is web site 29 only on pollution?

Line 177: which NOAA web site? (25?)

Line 179: ‘chlorophyll-a’, sea comment to line 95.

Line 179: which “Ocean color” resources?

Line 183: which “Planeta” resource (web site 23?)

Line 189: ‘sea state’ generally refer to waves. Do you mean ‘circulation’?

Line 194: change ‘some steps’ to ‘given resolution’ and ‘grid steps’ to ‘grid resolution’?

Line 198: ‘chlorophyll “a”’, sea comment to line 95.

Line 200: is it reference 25 or 26?

Line 201: change ‘stream’ to ‘river’?

Line 207: is ‘resort’ the correct word?

Line 217: ‘Open boundary’ is more commonly used than ‘Liquid boundary’. However, the latter is actually more precise.

Line 223: what do you mean by ‘urgent’? Perhaps ‘should be investigated’?

Line 226: what do you mean by ‘shifted towards’?

Line 230: add Voronka to the map, Figure 1.

Line 235: change ‘is a challenge’ to ‘demand high grid resolution’

Line 243: ‘Just a part..’ refer to Figure 1.

Line 248: who do you mean by ‘our’? add some more information on JASMINE; name and where it is developed.

Line 252: add full name of ‘BFM’

Line 256-257: change ‘horizons’ to ‘depth levels’ or ‘layers’?

Line 257: are the 31 levels 5 meters apart rather than 10 meters apart? These numbers do not add up to 300 meter bottom depth.

Line 259: which parts of the sea is closed for expeditions? You write ‘some parts’.

Line 262-264: your last sentence appear to me a bit out of the context of the paper.

Author Response

We agree with all suggestions and followed them. The text has been edited by an english-speaking colleague. The detailed report on the changes is attached as a PDF.

Round 2

Reviewer 1 Report

There is a problem with the English language, I have highlighted some, but I am not marking every error. I request the authors to check their manuscript for grammar and English by some person or use some software like Grammarly.

P1L15: “Pomors were rather isolated; they built special ships”.

P1L18: “Svalbard) and Novaya Zemlya archipelago in the Kara Sea”

P1L22: “remained important for fishing, essential for the economy of the country. Besides the fish, algae and molluscs were also harvested.”

P1L25: This sentence is hanging alone.

The Arctic is a noun; please use the Arctic, not arctic.

P9L299, Remote sensor (satellite) data can also be used. Why?

Author Response

There is a problem with the English language, I have highlighted some, but I am not marking every error. I request the authors to check their manuscript for grammar and English by some person or use some software like Grammarly.

-We have edited the manuscript with an expert in the English language and have processed it in Grammarly and onlinecorrection.com.

P1L15: “Pomors were rather isolated; they built special ships”.

-Corrected according to the advice

P1L18: “Svalbard) and Novaya Zemlya archipelago in the Kara Sea”

-Corrected according to the advice

P1L22: “remained important for fishing, essential for the economy of the country. Besides the fish, algae and molluscs were also harvested.”

-Corrected according to the advice

P1L25: This sentence is hanging alone.

-We decided to remove the sentence, because bathymetry is discussed below.

The Arctic is a noun; please use the Arctic, not arctic.

-Corrected dthroughout the text

P9L299, Remote sensor (satellite) data can also be used. Why?

-We rewrote the sentences: "Remote sensor (satellite) data is not always available due to cloudiness that prevails over the White Sea for half a year."

Reviewer 2 Report

I am happy with the changes made to the original manuscript.

Author Response

Thank you very much for your helpful review.